# Association of COVID-19 vaccines ChAdOx1 and BNT162b2 with major venous, arterial, or thrombocytopenic events: A population-based cohort study of 46 million adults in England

William N. Whiteley[1,2]*, Samantha Ip[3], Jennifer A. Cooper[4], Thomas Bolton[3,5], Spencer Keene[3], Venexia Walker[4], Rachel Denholm[4], Ashley Akbari[6], Efosa Omigie[7], Sam Hollings[7], Emanuele Di Angelantonio[3,8], Spiros Denaxas[5,9], Angela Wood[3,8☯], Jonathan A. C. Sterne[4,10,11☯], Cathie Sudlow[5☯], CVD-COVID-UK consortium

1 Centre for Clinical Brain Sciences, University of Edinburgh, Edinburgh, United Kingdom, 2 MRC Population Health Research Unit, Nuffield Department of Population Health University of Oxford, Oxford, United Kingdom, 3 Department of Public Health and Primary Care, University of Cambridge, Cambridge, United Kingdom, 4 Department of Population Health Sciences, University of Bristol, Bristol, United Kingdom, 5 BHF Data Science Centre, Health Data Research UK, London, United Kingdom, 6 Population Data Science, Health Data Research UK, Swansea University, Swansea, United Kingdom, 7 NHS Digital, Leeds, United Kingdom, 8 NIHR Blood and Transplant Research Unit in Donor Health and Genomics, University of Cambridge, Cambridge, United Kingdom, 9 Institute of Health Informatics, University College London, London, United Kingdom, 10 NIHR Bristol Biomedical Research Centre, Bristol, United Kingdom, 11 HDR UK South West, Bristol, United Kingdom

☯ These authors contributed equally to this work.
* william.whiteley@ed.ac.uk

## Abstract

### Background

Thromboses in unusual locations after the Coronavirus Disease 2019 (COVID-19) vaccine ChAdOx1-S have been reported, although their frequency with vaccines of different types is uncertain at a population level. The aim of this study was to estimate the population-level risks of hospitalised thrombocytopenia and major arterial and venous thromboses after COVID-19 vaccination.

### Methods and findings

In this whole-population cohort study, we analysed linked electronic health records from adults living in England, from 8 December 2020 to 18 March 2021. We estimated incidence rates and hazard ratios (HRs) for major arterial, venous, and thrombocytopenic outcomes 1 to 28 and >28 days after first vaccination dose for ChAdOx1-S and BNT162b2 vaccines. Analyses were performed separately for ages <70 and ≥70 years and adjusted for age, age[2], sex, ethnicity, and deprivation. We also prespecified adjustment for anticoagulant medication, combined oral contraceptive medication, hormone replacement therapy

**Data Availability Statement:** The de-identified data used in this study is available via the CVD-COVID-

UK consortium coordinated by BHF Data Science Centre for accredited researchers working on approved projects in the NHS Digital trusted research environment. For information on the access and approvals process, please email bhfdsc@hdruk.ac.uk or see https://web.www. healthdatagateway.org/dataset/7e5f0247-f033-4f98-aed3-3d7422b9dc6d.

**Funding:** The British Heart Foundation Data Science Centre (grant No SP/19/3/34678 awarded to Health Data Research (HDRUK) funded co-development (with NHS Digital) of the trusted research environment provision of linked datasets, data access, user software licences, computational usage, and data management and wrangling support. Support was also provided through the Data and Connectivity and Longitudinal Health and Wellbeing National Core Studies, which were established through the UK Government's Chief Scientific Adviser's National Core Studies program to coordinate COVID-19 priority research. Consortium partner organisations funded the time of contributing data analysts, biostatisticians, epidemiologists, and clinicians. WW is supported by the Chief Scientist's Office (CAF/01/17). CS, AW and WW are supported by Stroke Association (SA CV 20/100018). SI was funded by a BHF-Turing Cardiovascular Data Science 419 Award (BCDSA \100005) and is funded by the International Alliance for Cancer Early Detection, a partnership between Cancer Research UK C18081/A31373, Canary Center at Stanford University, the University of Cambridge, OHSU Knight Cancer Institute, University College London and the University of Manchester. AMW is supported by the BHF-Turing Cardiovascular Data Science Award (BCDSA/ 100005) and by core funding from UK MRC (MR/ L003120/1), BHF (RG/13/13/30194, RG/18/13/ 33946), and NIHR Cambridge Biomedical Research Centre (BRC/1215/20014). AMW and SD are part of the BigData@Heart Consortium, funded by the Innovative Medicines Initiative-2 Joint Undertaking under grant agreement No 116074. The views expressed are those of the authors and not necessarily those of the NIHR or the Department of Health and Social Care. SD is funded by the UCL Hospitals Biomedical Research Centre. JAC, JS, and RD are supported by the Health Data Research (HDR) UK South West Better Care Partnership, and the NIHR Bristol Biomedical Research Centre at University Hospitals Bristol, and Weston NHS Foundation Trust and the University of Bristol. VMW is supported by the MRC Integrative Epidemiology Unit, which receives its funding from the Medical Research Council and the University of Bristol (MC/UU/00011/4). AA, SD is supported by Health Data Research UK (grant number HDR/

medication, history of pulmonary embolism or deep vein thrombosis, and history of coronavirus infection in analyses of venous thrombosis; and diabetes, hypertension, smoking, antiplatelet medication, blood pressure lowering medication, lipid lowering medication, anticoagulant medication, history of stroke, and history of myocardial infarction in analyses of arterial thromboses. We selected further covariates with backward selection.

Of 46 million adults, 23 million (51%) were women; 39 million (84%) were <70; and 3.7 million (8.1%) Asian or Asian British, 1.6 million (3.5%) Black or Black British, 36 million (79%) White, 0.7 million (1.5%) mixed ethnicity, and 1.5 million (3.2%) were of another ethnicity. Approximately 21 million (46%) adults had their first vaccination between 8 December 2020 and 18 March 2021.

The crude incidence rates (per 100,000 person-years) of all venous events were as follows: prevaccination, 140 [95% confidence interval (CI): 138 to 142]; ≤28 days post-ChAdOx1-S, 294 (281 to 307); >28 days post-ChAdOx1-S, 359 (338 to 382), ≤28 days post-BNT162b2-S, 241 (229 to 253); >28 days post-BNT162b2-S 277 (263 to 291). The crude incidence rates (per 100,000 person-years) of all arterial events were as follows: prevaccination, 546 (95% CI: 541 to 555); ≤28 days post-ChAdOx1-S, 1,211 (1,185 to 1,237); >28 days post-ChAdOx1-S, 1678 (1,630 to 1,726), ≤28 days post-BNT162b2-S, 1,242 (1,214 to 1,269); >28 days post-BNT162b2-S, 1,539 (1,507 to 1,572).

Adjusted HRs (aHRs) 1 to 28 days after ChAdOx1-S, compared with unvaccinated rates, at ages <70 and ≥70 years, respectively, were 0.97 (95% CI: 0.90 to 1.05) and 0.58 (0.53 to 0.63) for venous thromboses, and 0.90 (0.86 to 0.95) and 0.76 (0.73 to 0.79) for arterial thromboses. Corresponding aHRs for BNT162b2 were 0.81 (0.74 to 0.88) and 0.57 (0.53 to 0.62) for venous thromboses, and 0.94 (0.90 to 0.99) and 0.72 (0.70 to 0.75) for arterial thromboses. aHRs for thrombotic events were higher at younger ages for venous thromboses after ChAdOx1-S, and for arterial thromboses after both vaccines.

Rates of intracranial venous thrombosis (ICVT) and of thrombocytopenia in adults aged <70 years were higher 1 to 28 days after ChAdOx1-S (aHRs 2.27, 95% CI: 1.33 to 3.88 and 1.71, 1.35 to 2.16, respectively), but not after BNT162b2 (0.59, 0.24 to 1.45 and 1.00, 0.75 to 1.34) compared with unvaccinated. The corresponding absolute excess risks of ICVT 1 to 28 days after ChAdOx1-S were 0.9 to 3 per million, varying by age and sex.

The main limitations of the study are as follows: (i) it relies on the accuracy of coded healthcare data to identify exposures, covariates, and outcomes; (ii) the use of primary reason for hospital admission to measure outcome, which improves the positive predictive value but may lead to an underestimation of incidence; and (iii) potential unmeasured confounding.

## Conclusions

In this study, we observed increases in rates of ICVT and thrombocytopenia after ChAdOx1-S vaccination in adults aged <70 years that were small compared with its effect in reducing COVID-19 morbidity and mortality, although more precise estimates for adults aged <40 years are needed. For people aged ≥70 years, rates of arterial or venous thrombotic events were generally lower after either vaccine compared with unvaccinated, suggesting that either vaccine is suitable in this age group.

9006), which receives its funding from the UK Medical Research Council, Engineering and Physical Sciences Research Council, Economic and Social Research Council, Department of Health and Social Care (England), Chief Scientist Office of the Scottish Government Health and Social Care Directorates, Health and Social Care Research and Development Division (Welsh Government), Public Health Agency (Northern Ireland), British Heart Foundation (BHF) and the Wellcome Trust, and Administrative Data Research UK, which is funded by the Economic and Social Research Council (grant number ES/S007393/1). The funders had no role in study design, data collection and analysis, decision to publish, or preparation of the manuscript.

**Competing interests:** I have read the journal's policy and the authors of this manuscript have the following competing interests: WW has given expert testimony to UK courts. WW served on a advisory board for Bayer. CS is Director of the BHF Data Science Centre (at Health Data Research UK), which worked with NHS Digital to develop its Trusted Research Environment. CS leads the CVD-COVID-UK consortium which has enabled access to the linked population health data that enabled this study. No other authors declared conflicts of interest.

**Abbreviations:** aHR, adjusted HR; BNF, British National Formulary; CI, confidence interval; COVID-19, Coronavirus Disease 2019; DIC, disseminated intravascular coagulation; DVT, deep vein thrombosis; EMA, European Medicine Agency; GDPPR, GPES Data for Pandemic Planning and Research; GPES, General Practice Extraction Service; HES, Hospital Episode Statistics; HR, hazard ratio; ICVT, intracranial venous thrombosis; MI, myocardial infarction; MHRA, UK Medicines and Healthcare Products Regulatory Agency; NHS, National Health Service; PE, pulmonary embolism; PF4, platelet factor 4; PRAC, Pharmacovigilance Risk Assessment Committee; SCCS, self-controlled case series; SNOMED-CT, Systematized Nomenclature of Medicine Clinical Terms; SUS, Secondary Uses Service; TTP, thrombotic thrombocytopenic purpura; VITT, vaccine-induced immune thrombotic thrombocytopenia.

## Author summary

### Why was this study done

- Cases of venous and arterial thromboses in unusual locations, such as the cerebral veins, and with low platelet levels, have been reported after vaccination with ChAdOx1-S COVID-19 vaccine.

- Case finding efforts in vaccinated people may lead to overestimation of risk associated with vaccines, if diagnostic thresholds vary between vaccinated and unvaccinated people.

- Effects of vaccination on rates of common venous and arterial events—such as stroke, myocardial infarction (MI), deep vein thrombosis (DVT), and pulmonary embolism (PE)—are difficult to measure based on case reports. Population-level data may give better estimates.

### What did the researchers do and find

- We used nationally collated data from electronic health records on 46 million adults, of whom 21 million were vaccinated during the study, and compared the incidence of venous and arterial events before and after the first vaccination with ChAdOx1-S and BNT162b2 COVID-19 vaccines.

- Overall rates of major arterial and venous events were lower after vaccination with both ChAdOx1-S and BNT162b2, after adjusting for potential confounding factors.

- In people <70 years, rates of hospitalisation due to intracranial venous thrombosis (ICVT) or due to thrombocytopenia were higher after vaccination with ChAdOx1-S but not BNT162b2, although the absolute increase in the risk of these events was very small.

### What do these findings mean

- In adults ≥70 years, a first vaccination with either ChAdOx1-S and BNT162b2 was not associated with an increase in rates major arterial or venous thrombotic events.

- In adults <70 years, the small increased risks of ICVT and hospitalisation with thrombocytopenia after first vaccination with ChAdOx1-S are likely to be outweighed by the vaccines' effect in reducing COVID-19 mortality and morbidity.

- The main limitations of the study were its reliance on the accuracy of coded electronic health records, and the potential for residual confounding.

## Introduction

In late February 2021, several groups reported a rare syndrome of thrombosis and thrombocytopenia after vaccination against Severe Acute Respiratory Syndrome Coronavirus-2 Disease 2019 (COVID-19) with ChAdOx1-S (developed by Oxford-AstraZeneca) [1–3]. Thromboses were found in unusual sites such as the cerebral venous sinuses, or mesenteric or portal veins. This syndrome, named "vaccine-induced immune thrombotic thrombocytopenia" (VITT), is probably due to an autoimmune response to platelet factor 4 (PF4) in the absence of exposure to heparin [1]. Whether the risk of common thrombotic illnesses is also increased postvaccination is uncertain, although there have been reports of postvaccination ischaemic stroke and intracerebral haemorrhage [1,3,4].

Estimates of vaccine-associated excess risk based on case reports and routine reporting may be biased if diagnostic thresholds vary between vaccinated and unvaccinated individuals or there is less than population-wide coverage. In England, the National Health Service (NHS) provides almost all healthcare; therefore, linked healthcare records provide comprehensive, population-wide data on outcomes before and after vaccination [5]. Vaccination in England began on 8 December 2020: The first groups vaccinated were care home residents, then those aged ≥80 years and frontline health and social care workers. Those clinically extremely vulnerable were invited alongside people aged ≥70 years, and those with other comorbidities aged <65 alongside people aged >65 years. Consequently, individuals vaccinated earlier are expected to have higher rates of venous and arterial events.

We conducted a new cohort study using routinely collected, linked health data from multiple sources, covering almost the entire English adult population (>46 million). We estimated associations of ChAdOx1-S and BNT162b2 with major venous and arterial events and thrombocytopenic haematological outcomes from the start of the vaccination programme to mid-March 2021, before VITT surveillance was widespread.

## Methods

Data were analysed according to a prespecified protocol and analysis plan that was published on GitHub on 5 May 2021 (S1 Protocol) [6]. We used the Strengthening the REporting of studies Conducted using Observational Routinely collected Data (RECORD) guideline to write this paper (S1 Record Checklist).

### Population

The study population was adults (aged ≥18 years), alive and registered with an English NHS general practice on 8 December 2020. The data resource [5] includes primary care data (General Practice Extraction Service (GPES) Data for Pandemic Planning and Research (GDPPR)) from 98% of general practices linked at individual-level to nationwide secondary care data including all NHS hospital admissions (Hospital Episode Statistics (HES) and Secondary Uses Service (SUS) data from 1997 onwards), COVID-19 laboratory testing data, COVID-19 vaccination data (NHS England Immunisation Management System), national community drug dispensing data (NHS BSA Dispensed Medicines from 2018), and death registrations. We accessed and analysed pseudonymised data within NHS Digital's secure, privacy-protecting Trusted Research Environment [7].

### Covariates

We defined covariates from primary care, hospital admissions, community drug dispensing, and COVID-19 laboratory testing data, using phenotyping algorithms verified by specialist

physicians (see study protocol). Phenotypes for comorbidities, risk factors, and other covariates used Systematized Nomenclature of Medicine Clinical Terms (SNOMED-CT) concepts for primary care data, and ICD-10 codes for hospital admission data. We defined sex, age, region, deprivation, and smoking status as the latest recorded in primary care records before 8 December 2020, and the most recently recorded ethnicity in either primary care or hospital admissions records was used. We defined history of diabetes, depression, obesity, cancer, chronic obstructive pulmonary disease, chronic kidney disease, stroke, myocardial infarction (MI), deep vein thrombosis (DVT) or pulmonary embolism (PE), cancer, thrombophilia, liver disease, or dementia as any record in primary care and/or hospital admission data before 8 December 2020. We derived the number of unique diseases in SNOMED-CT for the year before 8 December 2020 from primary care records and major surgery in the previous year from hospital admission records (using the Office of Population Censuses and Surveys classification of surgical procedures).

We defined a history of coronavirus (SARS-CoV-2) infection as either a positive COVID-19 antigen test in national laboratory data covering swab tests performed in the general population and hospitals or a confirmed COVID-19 diagnosis in primary care or hospital admission records. We defined prior medication using community dispensing data on all prescriptions dispensed by community pharmacists, appliance contractors, and doctors in England. British National Formulary (BNF) codes were used to define the total number of types of medication prescribed and dispensed in the following groups: antiplatelets, antihypertensives, lipid lowering agents, oral anticoagulants, combined oral contraceptives, and hormone replacement therapy.

## Outcomes

We derived outcomes from primary care data, hospital admission data (SUS dataset), and the national death registry (see S1 Protocol) [8]. We used specialist clinician-verified SNOMED-CT and ICD-10 rule-based phenotyping algorithms to define fatal or nonfatal (i) arterial thrombotic events: MI, ischaemic stroke (ischaemic or unclassified stroke, spinal stroke, or retinal infarction), other nonstroke non-MI arterial thromboembolism; (ii) venous thromboembolic events: PE, lower limb DVT, intracranial venous thrombosis (ICVT), portal vein thrombosis, and venous thrombosis at other sites; (iii) thrombocytopenic haematological events: any thrombocytopenia (idiopathic, primary, secondary, or unspecified), disseminated intravascular coagulation (DIC), and thrombotic thrombocytopenic purpura (TTP); (iv) death from any cause; (v) other vascular outcomes: haemorrhagic stroke (intracerebral or subarachnoid) and mesenteric thrombus for which available codes did not distinguish between arterial or venous causes; and (vi) lower limb fracture as a control condition unlikely to be affected by vaccination. The event date was the earliest of start of hospital episode, death, or recorded date of primary care event or consultation. We identified events within the death registry based on underlying cause of death and in hospital admission data based on the primary cause of the care episode.

## Statistical analyses

Follow-up was from 8 December 2020 to 18 March 2021, the date the European Medicine Agency (EMA) Pharmacovigilance Risk Assessment Committee (PRAC) discussed the first reported complications of vaccination, after which diagnostic effort was expected to be concentrated in people receiving ChAdOx1-S [9]. Any cell numbers <5 and any potentially disclosive numbers <10 are not reported exactly but as <5 and <10, respectively.

To allow for varying vaccination dates, we split follow-up from 8 December 2020 for each person into periods before, and 1 to 28, and >28 days after, first vaccination. Censoring was at the earliest of the outcome, death, 18 March 2021 and, in analyses of specific vaccines, receipt of the other vaccine. We estimated incidence rates per 100,000 people per year for each outcome, before and 1 to 28 and >28 days after vaccination.

We fitted Cox models with time zero 8 December 2020, ensuring that all analyses accounted for changes in rates of outcome events with calendar time. We estimated hazard ratios (HRs) comparing the periods 1 to 28 and >28 days after vaccination with unvaccinated or prevaccination person-time (reference). We fitted Cox models separately by age group (<70 and ≥70 years), both overall and separately for males and females. All Cox models were stratified by geographic region. For rare outcomes, sex-specific HRs were estimated from vaccine–sex interaction terms. For computational efficiency, each fitted model used data from all individuals with the outcome of interest and a random 10% sample of all other individuals; analyses incorporated inverse probability weights to account for this sampling [10,11]. We derived confidence intervals with robust standard errors.

Region-; age-sex-region-; and fully adjusted HRs (aHRs) for associations of ChAdOx1-S and BNT162b2 with outcome events were estimated. Analyses were adjusted for age, $age^2$, sex, ethnicity, deprivation, and relevant medical history. Further covariates were selected using backward selection, from models using MI as the outcome. Covariates had few missing values (apart from ethnicity, for which the 5.9% "missing" were included as a separate category); hence, all analyses used "complete-cases".

Sensitivity analyses examined outcome events recorded as the primary or secondary reason for admission or death in hospital admissions or death records; fatal outcome events (those followed by death from any cause within 28 days); and all venous and all arterial thrombotic outcome events associated with thrombocytopenia. Effect modification by age, sex, ethnicity, deprivation, medication, and medical history was examined for all venous and all arterial thromboses.

We calculated age- and sex-specific absolute excess risks of ICVT in days 1 to 28 following vaccination by multiplying estimated HRs by the average monthly incidence rates of first fatal or nonfatal ICVT; the latter were derived from records during calendar year 2019 in the population-wide data resource.

The study was approved by the Newcastle & North Tyneside 2 Research Ethics Committee (20/NE/0161), the NHS Digital Data Access Request Service (DARS-NIC-381078-Y9C5K), and the British Heart Foundation Data Science Centre CVD-COVID-UK Approvals and Oversight Board.

Information on the data used can be found on the HDR UK Gateway https://web.www.healthdatagateway.org/dataset/7e5f0247-f033-4f98-aed3-3d7422b9dc6d; EHR phenotyping algorithms can be downloaded in machine-readable formats from the HDR UK Phenotype Library http://phenotypes.healthdatagateway.org. Data manipulation and analyses used SQL and Python in Databricks and RStudio (Professional) Version 1.3.1093.1 driven by R Version 4.0.3. All code and phenotypes are available at github.com/BHFDSC/CCU002_02.

### Deviations from prespecified protocol

In this paper, we concentrate on the effects of first COVID-19 vaccination on major thrombotic events. The results of analyses on thrombotic events after COVID infection and second vaccination will be presented in future papers.

### Results

On 8 December 2020, 46,162,942 eligible adults were registered with an English general practice. By 18 March 2021, 21,193,814 (46%) had received their first vaccination (8,712,477 BNT162b2; 12,481,337 ChAdOx1-S) (S1 Fig). Postvaccination person-years of follow-up for

ages <40, 40 to 69, and ≥70 years, respectively, were for ChAdOx1-S: 96,745, 455,122, and 418,942 person-years; for BNT162b2: 165,815, 466,025, and 603,436 person-years. Total unvaccinated follow-up time for ages <40, 40 to 69, and ≥70 years was 4,529,022, 5,026,046, and 1,008,498 person-years, respectively. The median follow-up per person was 101 days.

The risks per 100,000 persons from 8 December 2020 to 18 March 2021 were for the following: any venous thrombosis 45.3; any arterial thrombosis 189; thrombocytopenia 4.2 (Table 1, age-stratified estimates S1 Table). Risks of venous and arterial thromboses were higher in people with comorbidities and with increasing age and deprivation, and varied substantially by ethnicity. Absolute and relative increases in risk with increasing age and deprivation were greater for arterial than for venous thromboses. The risk per 100,000 people during follow-up of venous thromboses was higher in people with prior DVT or PE (541), thrombophilia (327), or oral anticoagulant medication (170), while the risk of arterial thromboses was higher in people with prior stroke (2,955), MI (2,036), or antiplatelet medication (1,451). The risk of thrombocytopenia increased with age and comorbidities but varied less markedly with ethnicity and deprivation than the risk of venous or arterial thromboses.

The additional confounding factors selected using backward selection were as follows: for both sexes and age strata, previous diagnosis of cancer, number of unique diseases in the last year, surgery in the last year, obesity, liver disease; additionally for women aged <70, history of depression, and total number of types of medication (by BNF chapters); additionally for men aged <70, history of SARS-CoV-2 infection and thrombocytopenia; and additionally for men and women aged ≥70, chronic kidney disease, and dementia.

The crude incidence rates of all events varied substantially in unvaccinated and postvaccination (Tables 2 and 3 age-stratified estimates S1 Table), and the crude HRs comparing incidence rates after with before vaccination were substantially attenuated after adjusting for age and further attenuated after adjusting for confounding factors. For example, in <70-year-olds, the HRs 1 to 28 days after vaccination with ChAdOx1-S versus unvaccinated were attenuated from 2.81 (unadjusted) to 2.27 (adjusted) for ICVT; from 4.85 to 1.71 for thrombocytopenia; and from 3.21 to 0.90 for ischaemic stroke (Tables 4 and 5).

In people aged <70 years, the fully adjusted HR for any venous thrombosis was 0.97 (95% confidence interval (CI): 0.90 to 1.05) 1 to 28 days after ChAdOx1-S compared with before vaccination, with similar aHRs for PE and DVT. Compared with before vaccination, the hazard of any venous thrombosis was lower 1 to 28 days after ChAdOx1-S in people aged ≥70 (aHR 0.58, 95% CI: 0.53 to 0.63) and 1 to 28 days after BNT162b2 (<70: HR 0.81, 0.74 to 0.88; ≥70 HR: 0.57, 0.53 to 0.62), with similar aHRs for PE and DVT. (Fig 1, Tables 4 and 5).

In people aged <70 years, aHRs for ICVT were 2.27 (95% CI: 1.32 to 3.88) 1 to 28 days after ChAdOx1-S and 0.59 (0.24 to 1.45) 1 to 28 days after BNT162b2 compared with before vaccination. HRs were similar in men and women (Fig 1, Tables 4 and 5). In a post hoc analyses, in people aged <40 years, the aHRs for ICVT 1 to 28 days after vaccination were 3.73 (1.58 to 8.83) and 1.07 (0.33 to 3.48) for ChAdOx1-S and BNT162b2, respectively, compared with before vaccination. The same estimates for people 40 to 69 years were 1.93 (1.19 to 2.13) and 0.74 (0.29 to 1.87). In people aged ≥70 years, associations of vaccination with ICVT were estimated imprecisely (<10 events for ChAdOx1-S and 8 for BNT162b2).

aHRs for any arterial thrombosis 1 to 28 days after ChAdOx1-S were 0.90 (95% CI: 0.86 to 0.95) and 0.76 (0.73 to 0.79) in people aged <70 and ≥70 years, respectively, compared with before vaccination. Corresponding aHRs 1 to 28 days after BNT162b2 were 0.94 (0.90 to 0.99) and 0.72 (0.70 to 0.75). HRs were similar for MI and ischaemic stroke, and >28 days after vaccination (Fig 2, Tables 4 and 5).

In people aged <70 years, aHRs for thrombocytopenia were 1.71 (95% CI:1.35 to 2.16) 1 to 28 days and 1.69 (1.16 to 2.46) >28 days after ChAdOx1-S compared with before vaccination.

**Table 1. Numbers of patients analysed and (in parentheses) risk per 100,000 people for the whole population during follow-up of venous and arterial events and TCP.**

| | | Overall | | | |
|---|---|---|---|---|---|
| | | **All** | **Venous** | **Arterial** | **TCP** |
| N | | 46,162,942 | 20,903 (45.3) | 87,251 (189) | 1,926 (4.2) |
| Sex | Male | 22,765,779 | 10,358 (45.5) | 51,851 (228) | 986 (4.3) |
| | Female | 23,397,163 | 10,545 (45.1) | 35,400 (151) | 940 (4.0) |
| Age | 18–29 | 8,821,973 | 781 (8.9) | 400 (4.5) | 143 (1.6) |
| | 30–49 | 16,131,025 | 3,544 (22.0) | 6,424 (39.8) | 338 (2.1) |
| | 50–69 | 13,791,173 | 7,510 (54.5) | 31,459 (228) | 686 (5.0) |
| | 70–79 | 4,674,973 | 4,820 (103) | 22,249 (476) | 418 (8.9) |
| | 80+ | 2,743,798 | 4,248 (155) | 26,719 (974) | 341 (12.4) |
| Ethnicity | Asian or Asian British | 3,718,442 | 599 (16.1) | 5,325 (143) | 152 (4.1) |
| | Black or Black British | 1,634,470 | 654 (40.0) | 2,086 (128) | 58 (3.5) |
| | Mixed | 712,534 | 182 (25.5) | 642 (90.1) | 22 (3.1) |
| | Other ethnic groups | 1,497,133 | 217 (14.5) | 1,072 (71.6) | 39 (2.6) |
| | White | 36,446,855 | 18,995 (52.1) | 77,118 (212) | 1,643 (4.5) |
| | Unknown or missing | 2,153,508 | 256 (11.9) | 1,008 (46.8) | 12 (0.6) |
| Deprivation[1] | 1–2 | 9,118,746 | 4,565 (50.1) | 18,830 (206) | 365 (4.0) |
| | 3–4 | 9,567,633 | 4,105 (42.9) | 17,436 (182) | 370 (3.9) |
| | 5–6 | 9,355,549 | 4,091 (43.7) | 17,642 (189) | 392 (4.2) |
| | 7–8 | 9,076,313 | 4,146 (45.7) | 16,939 (187) | 381 (4.2) |
| | 9–10 | 8,772,902 | 3,926 (44.8) | 16,024 (183) | 406 (4.6) |
| Smoking status | Current | 7,814,045 | 3,079 (39.4) | 16,115 (206) | 260 (3.3) |
| | Former | 10,623,072 | 6,930 (65.2) | 31,768 (299) | 581 (5.5) |
| | Never | 25,722,494 | 10,667 (41.5) | 38,701 (150) | 1,053 (4.1) |
| Medical history | Stroke | 727,218 | 1,020 (140) | 21,489 (2,955) | 104 (14.3) |
| | MI | 1,189,182 | 1,436 (121) | 24,212 (2,036) | 158 (13.3) |
| | DVT or PE | 603,351 | 3,263 (541) | 4,050 (671) | 142 (23.5) |
| | Thrombophilia | 44,593 | 146 (327) | 202 (453) | 23 (51.6) |
| | Coronavirus infection[2] | 1,284,984 | 1,131 (88.0) | 4,167 (324) | 101 (7.9) |
| | Diabetes | 4,069,412 | 3,654 (89.8) | 25,595 (629) | 423 (10.4) |
| | Depression | 9,516,020 | 6,272 (65.9) | 24,146 (254) | 500 (5.3) |
| | Obesity | 10,918,274 | 8,863 (81.2) | 31,194 (286) | 679 (6.2) |
| | Cancer | 6,609,691 | 5,674 (85.8) | 17,827 (270) | 786 (11.9) |
| | COPD | 1,582,366 | 2,497 (158) | 12,323 (779) | 182 (11.5) |
| | Liver disease | 230,243 | 284 (123) | 1,069 (464) | 97 (42.1) |
| | CKD | 2,934,107 | 4,467 (152) | 25,816 (880) | 526 (17.9) |
| | Major surgery[3] | 3,995,870 | 5,407 (135) | 24,164 (605) | 763 (19.1) |
| | Dementia | 524,293 | 1,127 (215) | 6,073 (1,158) | 45 (8.6) |
| Medications | Antiplatelet | 2,510,382 | 2,814 (112) | 36,415 (1,451) | 218 (8.7) |
| | BP lowering | 8,589,860 | 7,969 (92.8) | 55,462 (646) | 777 (9.0) |
| | Lipid lowering | 6,808,408 | 5,796 (85.1) | 47,859 (703) | 564 (8.3) |
| | Anticoagulant | 1,338,585 | 2,278 (170) | 12,758 (953) | 205 (15.3) |
| | Oral contraceptive | 622,529 | 138 (22.2) | 42 (6.7) | 8 (1.3) |
| | HRT | 540,722 | 213 (39.4) | 516 (95.4) | 16 (3.0) |
| Number of diagnoses | 0 | 36,455,307 | 12,200 (33.5) | 38,617 (106) | 980 (2.7) |
| | 1–5 | 9,573,068 | 8,422 (88.0) | 46,012 (481) | 925 (9.7) |
| | 6+ | 134,567 | 281 (209) | 2,622 (1948) | 21 (15.6) |

*(Continued)*

**Table 1.** (Continued)

| | | Overall | | | |
|---|---|---|---|---|---|
| | | **All** | **Venous** | **Arterial** | **TCP** |
| Number of medications | 0 | 22,970,920 | 3,679 (16.0) | 9,629 (41.9) | 321 (1.4) |
| | 1–5 | 20,875,516 | 12,872 (61.7) | 56,255 (269) | 1,210 (5.8) |
| | 6+ | 2,316,506 | 4,352 (188) | 21,367 (922) | 395 (17.1) |

[1]Index of Multiple Deprivation deciles where 10 indicates least deprived and 1 indicates most deprived.

[2]After 31 December 2019 and prior to 8 December 2020.

[3]In the last year.

BP, blood pressure; CKD, chronic kidney disease; COPD, chronic obstructive respiratory disease; DVT, deep vein thrombosis; HRT, hormone replacement therapy; MI, myocardial infarction; PE, pulmonary embolism; TCP, thrombocytopenia.

Corresponding HRs after BNT162b2 were 1.00 (0.75 to 1.34) and 0.97 (0.66 to 1.41). For those aged ≥70 years, aHRs for thrombocytopenia 1 to 28 days after vaccination were similar for ChAdOx1 (0.79, 95% CI: 0.56 to 1.10) and BNT162b2 (0.68, 0.51 to 0.90) (Fig 3, Tables 4 and 5).

aHRs for the control condition lower limb fracture 1 to 28 days after ChAdOx1-S were 0.85 (95% CI: 0.77 to 0.93), and 0.83 (0.78 to 0.89) in people aged <70 and ≥70 years, respectively, compared with before vaccination. Corresponding aHRs after BNT162b2 were 0.93 (0.84 to

**Table 2. Numbers and incidence rates pre- and post-first ChAdOx1-S.** Incidence rate per 100,000 person years. Disclosure control prevents presentation of $n > 5$.

| Outcome | Prevaccination or unvaccinated | | Postvaccination ≤28 days | | Postvaccination >28 days | |
|---|---|---|---|---|---|---|
| | N (events) | Incidence rate (95% CI) | N (events) | Incidence rate (95% CI) | N (events) | Incidence rate (95% CI) |
| **All venous** | 14,769 | 139.8 (137.6–142.1) | 2,006 | 293.6 (281.0–306.7) | 995 | 359.3 (337.6–382.3) |
| ICVT | 207 | 1.96 (1.71–2.25) | 27 | 3.95 (2.71–5.76) | 9 | 3.25 (1.69–6.24) |
| Portal vein thrombosis | 121 | 1.15 (0.96–1.37) | 14 | 2.05 (1.21–3.46) | 9 | 3.25 (1.69–6.24) |
| PE | 8,473 | 80.2 (78.5–81.9) | 1,203 | 176.0 (166.4–186.3) | 587 | 211.9 (195.4–229.7) |
| DVT | 5,664 | 53.6 (52.2–55.0) | 739 | 108.1 (100.6–116.2) | 374 | 135.0 (122.0–149.4) |
| Other | 459 | 4.35 (3.97–4.76) | 51 | 7.46 (5.67–9.82) | 30 | 10.8 (7.6–15.5) |
| **All arterial** | 57,602 | 545.6 (541.1–550.0) | 8,256 | 1,210.6 (1184.8–1237.0) | 4,634 | 1,677.8 (1630.2–1726.8) |
| MI | 28,134 | 266.4 (263.3–269.5) | 3,814 | 558.5 (541.1–576.5) | 2,050 | 740.7 (709.4–773.5) |
| Ischaemic stroke (ischaemic, unknown, spinal) | 28,639 | 271.2 (268.1–274.3) | 4,334 | 634.8 (616.1–654.0) | 2,539 | 917.9 (882.9–954.3) |
| Other arterial | 1,308 | 12.4 (11.7–13.1) | 189 | 27.6 (24.0–31.9) | 98 | 35.4 (29.0–43.1) |
| **Haematological** | | | | | | |
| DIC | 13 | 0.12 (0.07–0.21) | <5 | 0.44 (0.14–1.36) | <5 | - |
| TTP | 70 | 0.66 (0.52–0.84) | <5 | 0.59 (0.22–1.56) | <5 | 0.72 (0.18–2.89) |
| Any thrombocytopenia | 1,357 | 12.8 (12.2–13.5) | 200 | 29.3 (25.5–33.6) | 89 | 32.1 (26.1–39.5) |
| **Other** | | | | | | |
| Haemorrhagic stroke (intracerebral or subarachnoid) | 3,747 | 35.5 (34.4–36.6) | 487 | 71.2 (65.2–77.9) | 289 | 104.3 (92.9–117.0) |
| Mesenteric thrombosis | 1,170 | 11.1 (10.5–11.7) | 171 | 25.0 (21.5–29.1) | 109 | 39.3 (32.6–47.5) |
| Lower limb fracture | 18,347 | 173.7 (171.2–176.3) | 2,657 | 389.0 (374.5–404.0) | 1,632 | 589.7 (561.7–619.0) |
| Death | 123,580 | 1,169.9 (1163.4–1176.4) | 16,192 | 2,368.5 (2332.3–2405.2) | 11,738 | 4,235.1 (4159.2–4312.4) |

DIC, disseminated intravascular coagulation; DVT, deep vein thrombosis; ICVT, intracranial venous thrombosis; MI, myocardial infarction; PE, pulmonary embolism; TTP, thrombotic thrombocytopenic purpura.

**Table 3. Numbers and incidence rates pre- and post-first BNT162b2. Incidence rate per 100,000 person years.** Disclosure control prevents presentation of $n > 5$.

| Outcome | Prevaccination or unvaccinated | | Postvaccination ≤28 days | | Postvaccination >28 days | |
|---|---|---|---|---|---|---|
| | N (events) | Incidence rate (95% CI) | N (events) | Incidence rate (95% CI) | N (events) | Incidence rate (95% CI) |
| **All venous** | 14,769 | 139.8 (137.6–142.1) | 1,546 | 240.8 (229.1–253.1) | 1,587 | 277.1 (263.8–291.0) |
| ICVT | 207 | 1.96 (1.71–2.25) | 13 | 2.02 (1.18–3.49) | 6 | 1.05 (0.47–2.33) |
| Portal vein thrombosis | 121 | 1.15 (0.96–1.37) | 5 | 0.78 (0.32–1.87) | 12 | 2.09 (1.19–3.69) |
| PE | 8,473 | 80.2 (78.5–81.9) | 928 | 144.5 (135.5–154.1) | 955 | 166.7 (156.5–177.6) |
| DVT | 5,664 | 53.6 (52.2–55.0) | 555 | 86.4 (79.5–93.9) | 590 | 103.0 (95.0–111.6) |
| Other | 459 | 4.35 (3.97–4.76) | 55 | 8.56 (6.58–11.15) | 39 | 6.81 (4.97–9.31) |
| **All arterial** | 57,602 | 545.6 (541.1–550.0) | 7,960 | 1,241.8 (1214.9–1269.4) | 8,799 | 1,539.7 (1507.8–1572.2) |
| MI | 28,134 | 266.4 (263.3–269.5) | 3,722 | 580.1 (561.8–599.0) | 4,023 | 702.9 (681.5–724.9) |
| Ischaemic stroke (ischaemic, unknown, spinal) | 28,639 | 271.2 (268.1–274.3) | 4,143 | 645.7 (626.4–665.7) | 4,702 | 821.8 (798.6–845.6) |
| Other arterial | 1,308 | 12.4 (11.7–13.1) | 156 | 24.3 (20.8–28.4) | 167 | 29.1 (25.0–33.9) |
| **Haematological** | | | | | | |
| DIC | 13 | 0.12 (0.07–0.21) | <5 | 0.31 (0.08–1.25) | <5 | 0.35 (0.09–1.40) |
| TTP | 70 | 0.66 (0.52–0.84) | <5 | 0.62 (0.23–1.66) | <5 | 0.35 (0.09–1.40) |
| Any thrombocytopenia | 1,357 | 12.8 (12.2–13.5) | 136 | 21.2 (17.9–25.1) | 144 | 25.1 (21.3–29.6) |
| **Other** | | | | | | |
| Haemorrhagic stroke (intracerebral or subarachnoid) | 3,747 | 35.5 (34.4–36.6) | 440 | 68.5 (62.4–75.2) | 579 | 101.0 (93.1–109.6) |
| Mesenteric thrombosis | 1,170 | 11.1 (10.5–11.7) | 161 | 25.1 (21.5–29.3) | 217 | 37.9 (33.2–43.3) |
| Lower limb fracture | 18,347 | 173.7 (171.2–176.3) | 2,573 | 400.8 (385.6–416.6) | 3,176 | 554.7 (535.8–574.4) |
| Death | 123,580 | 1,169.9 (1163.4–1176.4) | 9,117 | 1,419.6 (1390.8–1449.0) | 13,935 | 2,431.7 (2391.7–2472.4) |

CI, confidence interval; DIC, disseminated intravascular coagulation; DVT, deep vein thrombosis; ICVT, intracranial venous thrombosis; MI, myocardial infarction; PE, pulmonary embolism; TTP, thrombotic thrombocytopenic purpura.

1.02) and 0.69 (0.66 to 0.73). (Fig 3, Tables 4 and 5). aHRs for all-cause mortality 1 to 28 days after ChAdOx1-S were 0.37 (95% CI: 0.35 to 0.39), and 0.28 (0.28 to 0.29) in people aged <70 and ≥70 years, respectively. Corresponding aHRs after BNT162b2 were 0.24 (0.22 to 0.26) and 0.19 (0.19 to 0.20) (S2 Fig, Tables 4 and 5).

## Subgroup analyses

aHRs for all venous and for all arterial thrombotic events 1 to 28 days after vaccination compared with before vaccination were lower in older age groups for each vaccine (all interaction *p*-values <0.001) (S2 Table). HRs for venous thromboses were >1 1 to 28 days after compared with before vaccination in people aged <50 years for ChAdOx1-S but not BNT162b2. aHRs for arterial thromboses were >1 1 to 28 days after compared with before vaccination in people aged <50 years for both ChAdOx1-S and BNT162b2. Less marked trends for lower HRs in older individuals were observed for HRs >28 days after vaccination. A higher proportion of venous thromboses in those aged <40 years (7%) compared with those aged ≥40 years were due to ICVT (40 to 59: 1.6%, 60 to 69: 0.9%, and 70+: 0.5%). A higher proportion of arterial thromboses at age <40 (49%) was due to stroke than at ages 40 to 59 (40%) and 60 to 69 (44%) years.

Despite modest differences between the magnitude of HRs in subgroups defined by sex, ethnicity, prior COVID, COCP, diabetes, deprivation or history of MI, DVT, or PE, anticoagulation or antiplatelet medication, aHRs in all of these subgroups were consistent with lower

**Table 4. HRs (95% CIs) for thrombotic and other outcomes 1–28 and >28 days postvaccination with ChAdOx1-S vaccine, compared with prevaccination rates.** All analyses were stratified on geographical region.

| Outcome | Age | 1–28 days postvaccination | | | >28 days postvaccination | | |
|---|---|---|---|---|---|---|---|
| | | Unadjusted[1] | Age and sex adjusted[2] | Fully adjusted[3] | Unadjusted[1] | Age and sex adjusted[2] | Fully adjusted[3] |
| **All venous events** | <70 | 2.33 (2.16–2.51) | 1.32 (1.22–1.43) | 0.97 (0.90–1.05) | 2.74 (2.39–3.14) | 1.60 (1.38–1.84) | 0.94 (0.81–1.08) |
| | ≥70 | 0.65 (0.59–0.72) | 0.64 (0.59–0.70) | 0.58 (0.53–0.63) | 0.67 (0.59–0.76) | 0.58 (0.52–0.65) | 0.49 (0.44–0.55) |
| ICVT | <70 | 2.81 (1.70–4.64) | 2.99 (1.75–5.09) | 2.27 (1.33–3.88) | 2.37 (0.91–6.12) | 2.47 (0.95–6.42) | 1.72 (0.66–4.49) |
| | ≥70 | 0.62 (0.17–2.23) | 0.65 (0.19–2.21) | 0.67 (0.20–2.18) | 1.35 (0.13–13.46) | 1.13 (0.14–9.20) | 1.11 (0.14–9.02) |
| Portal vein thrombosis | <70 | 2.49 (1.20–5.14) | 1.56 (0.72–3.40) | 1.00 (0.24–4.15) | 2.76 (0.73–10.51) | 1.99 (0.53–7.52) | 1.00 (0.06–18.18) |
| | ≥70 | 0.66 (0.11–4.06) | 0.67 (0.10–4.24) | 0.61 (0.10–3.71) | 2.72 (0.49–15.02) | 3.00 (0.48–18.99) | 2.61 (0.41–16.52) |
| PE | <70 | 2.54 (2.30–2.80) | 1.28 (1.15–1.43) | 0.95 (0.85–1.05) | 3.12 (2.61–3.73) | 1.63 (1.34–1.96) | 0.95 (0.79–1.15) |
| | ≥70 | 0.62 (0.55–0.70) | 0.60 (0.54–0.68) | 0.54 (0.48–0.61) | 0.61 (0.52–0.72) | 0.54 (0.46–0.62) | 0.45 (0.39–0.53) |
| DVT | <70 | 2.08 (1.85–2.34) | 1.34 (1.18–1.52) | 0.99 (0.87–1.12) | 2.37 (1.90–2.96) | 1.58 (1.25–1.99) | 0.97 (0.77–1.22) |
| | ≥70 | 0.73 (0.62–0.86) | 0.71 (0.61–0.83) | 0.63 (0.54–0.74) | 0.71 (0.57–0.88) | 0.62 (0.51–0.75) | 0.51 (0.42–0.62) |
| Other | <70 | 2.48 (1.70–3.62) | 1.60 (1.06–2.40) | 1.02 (0.69–1.52) | 3.04 (1.54–6.00) | 2.04 (1.01–4.09) | 0.94 (0.47–1.91) |
| | ≥70 | 0.46 (0.21–1.01) | 0.46 (0.21–0.99) | 0.43 (0.20–0.92) | 1.13 (0.50–2.54) | 1.03 (0.48–2.21) | 0.93 (0.42–2.03) |
| **All arterial events** | <70 | 3.02 (2.90–3.14) | 1.25 (1.20–1.31) | 0.90 (0.86–0.95) | 3.55 (3.29–3.83) | 1.57 (1.45–1.70) | 0.91 (0.84–1.00) |
| | ≥70 | 0.89 (0.85–0.93) | 0.84 (0.80–0.87) | 0.76 (0.73–0.79) | 1.03 (0.97–1.10) | 0.83 (0.78–0.87) | 0.72 (0.68–0.77) |
| MI | <70 | 2.85 (2.70–3.01) | 1.24 (1.17–1.32) | 0.88 (0.83–0.94) | 2.93 (2.63–3.26) | 1.40 (1.24–1.56) | 0.83 (0.73–0.93) |
| | ≥70 | 0.86 (0.80–0.92) | 0.83 (0.78–0.88) | 0.76 (0.71–0.81) | 0.97 (0.89–1.07) | 0.83 (0.76–0.90) | 0.74 (0.68–0.81) |
| Ischaemic stroke[4] | <70 | 3.21 (3.02–3.41) | 1.25 (1.17–1.34) | 0.90 (0.84–0.96) | 4.36 (3.91–4.86) | 1.76 (1.56–1.98) | 0.94 (0.84–1.07) |
| | ≥70 | 0.92 (0.87–0.98) | 0.85 (0.81–0.90) | 0.77 (0.73–0.82) | 1.08 (0.99–1.18) | 0.83 (0.77–0.90) | 0.72 (0.67–0.78) |
| Other arterial | <70 | 3.18 (2.48–4.07) | 1.24 (0.93–1.64) | 0.82 (0.63–1.07) | 4.20 (2.62–6.73) | 1.68 (0.99–2.86) | 0.77 (0.46–1.30) |
| | ≥70 | 0.44 (0.33–0.59) | 0.45 (0.34–0.59) | 0.41 (0.31–0.53) | 0.55 (0.38–0.79) | 0.49 (0.35–0.70) | 0.44 (0.31–0.63) |
| **Haematological events** | | | | | | | |
| Any thrombocytopenia | <70 | 4.85 (3.87–6.07) | 3.00 (2.34–3.85) | 1.71 (1.35–2.16) | 7.48 (5.31–10.53) | 4.68 (3.24–6.77) | 1.69 (1.16–2.46) |
| | ≥70 | 0.92 (0.65–1.29) | 0.87 (0.63–1.22) | 0.79 (0.56–1.10) | 1.14 (0.66–1.95) | 0.99 (0.59–1.65) | 0.84 (0.50–1.41) |
| **Other events** | | | | | | | |
| Haemorrhagic stroke[5] | <70 | 2.25 (1.88–2.69) | 1.05 (0.86–1.27) | 1.00 (0.73–1.37) | 2.87 (2.10–3.92) | 1.39 (1.00–1.93) | 1.00 (0.51–1.95) |
| | ≥70 | 0.83 (0.69–1.00) | 0.78 (0.66–0.92) | 0.73 (0.62–0.87) | 1.09 (0.84–1.40) | 0.84 (0.67–1.04) | 0.76 (0.61–0.95) |
| Mesenteric thrombosis | <70 | 3.25 (2.42–4.37) | 1.24 (0.89–1.73) | 0.84 (0.61–1.16) | 5.47 (3.33–8.99) | 2.16 (1.24–3.77) | 1.04 (0.60–1.80) |
| | ≥70 | 0.66 (0.49–0.89) | 0.60 (0.46–0.79) | 0.53 (0.41–0.70) | 0.95 (0.64–1.43) | 0.74 (0.52–1.05) | 0.62 (0.43–0.89) |
| Lower limb fracture | <70 | 1.71 (1.55–1.87) | 1.01 (0.91–1.11) | 0.85 (0.77–0.93) | 2.26 (1.91–2.67) | 1.35 (1.13–1.60) | 0.98 (0.83–1.17) |
| | ≥70 | 1.11 (1.02–1.21) | 0.94 (0.88–1.01) | 0.83 (0.78–0.89) | 1.56 (1.39–1.76) | 0.95 (0.87–1.04) | 0.78 (0.71–0.85) |
| Death | <70 | 1.96 (1.87–2.06) | 0.60 (0.57–0.63) | 0.37 (0.35–0.39) | 4.51 (4.20–4.85) | 1.41 (1.30–1.53) | 0.51 (0.47–0.56) |
| | ≥70 | 0.35 (0.34–0.36) | 0.34 (0.33–0.35) | 0.28 (0.28–0.29) | 0.49 (0.47–0.51) | 0.38 (0.37–0.39) | 0.27 (0.26–0.28) |

[1]Stratifed by region.

[2]Adjusted for age, age², sex, and stratified by region.

[3]Adjusted for age, age², sex, ethnicity, deprivation, smoking, previous diagnosis of cancer, number of unique diseases in the last year, surgery in the last year, obesity, liver disease; additionally for women aged <70, history of depression and total number of types of medication (by BNF chapters); additionally for men aged <70, history of SARS-CoV2 infection and thrombocytopenia; and additionally for men and women aged ≥70, chronic kidney disease and dementia. Venous events adjusted for: anticoagulant medication, combined oral contraceptive medication, hormone replacement therapy medication, history of PE or DVT, and history of coronavirus infection. Arterial thromboses adjusted for: diabetes, hypertension, smoking, antiplatelet medication, blood pressure lowering medication, lipid lowering medication, anticoagulant medication, history of stroke, and history of MI. All analyses stratified by region.

[4]Ischaemic stroke: including ischaemic stroke, stroke of uncertain cause, and spinal stroke.

[5]Haemorrhagic stroke: including intracranial haemorrhage and subarachnoid haemorrhage.

BNF, British National Formulary; CI, confidence interval; DVT, deep vein thrombosis; HR, hazard ratio; ICVT, intracranial venous thrombosis; MI, myocardial infarction; PE, pulmonary embolism; SARS-CoV-2, Severe Acute Respiratory Syndrome Coronavirus 2.

**Table 5. HRs (95% CIs) for thrombotic and other outcomes 1–28 and >28 days postvaccination with BNT162b2 vaccine, compared with prevaccination rates.** All analyses were stratified on geographical region.

| Outcome | Age | 1–28 days postvaccination | | | >28 days postvaccination | | |
|---|---|---|---|---|---|---|---|
| | | Unadjusted[1] | Age and sex adjusted[2] | Fully adjusted[3] | Unadjusted[1] | Age and sex adjusted[2] | Fully adjusted[3] |
| **All venous events** | <70 | 1.47 (1.35–1.61) | 1.01 (0.93–1.11) | 0.81 (0.74–0.88) | 1.14 (1.01–1.29) | 0.88 (0.77–1.00) | 0.73 (0.64–0.83) |
| | ≥70 | 0.65 (0.60–0.71) | 0.56 (0.52–0.61) | 0.57 (0.53–0.62) | 0.68 (0.62–0.75) | 0.50 (0.46–0.55) | 0.50 (0.45–0.55) |
| ICVT | <70 | 0.76 (0.31–1.85) | 0.72 (0.29–1.78) | 0.59 (0.24–1.45) | 0.65 (0.20–2.13) | 0.59 (0.18–1.93) | 0.51 (0.16–1.68) |
| | ≥70 | 1.65 (0.64–4.29) | 1.25 (0.48–3.26) | 1.43 (0.55–3.75) | 1.13 (0.13–9.75) | 0.71 (0.08–6.56) | 0.88 (0.10–7.52) |
| Portal vein thrombosis | <70 | 0.58 (0.15–2.30) | 0.44 (0.11–1.72) | 0.29 (0.07–1.17) | 3.06 (1.29–7.24) | 2.84 (1.19–6.74) | 2.19 (0.89–5.37) |
| | ≥70 | 0.49 (0.10–2.28) | 0.43 (0.10–1.87) | 0.42 (0.09–1.91) | 0.67 (0.23–1.97) | 0.54 (0.19–1.56) | 0.48 (0.17–1.38) |
| PE | <70 | 1.52 (1.36–1.71) | 0.99 (0.87–1.11) | 0.78 (0.69–0.88) | 1.20 (1.02–1.43) | 0.89 (0.75–1.06) | 0.71 (0.60–0.85) |
| | ≥70 | 0.61 (0.55–0.68) | 0.53 (0.48–0.59) | 0.54 (0.49–0.60) | 0.61 (0.54–0.69) | 0.45 (0.40–0.51) | 0.45 (0.39–0.50) |
| DVT | <70 | 1.38 (1.21–1.59) | 1.03 (0.89–1.18) | 0.82 (0.71–0.95) | 1.05 (0.86–1.28) | 0.85 (0.69–1.04) | 0.73 (0.59–0.89) |
| | ≥70 | 0.69 (0.60–0.79) | 0.58 (0.51–0.67) | 0.61 (0.53–0.70) | 0.81 (0.69–0.96) | 0.59 (0.50–0.70) | 0.61 (0.52–0.73) |
| Other | <70 | 2.16 (1.47–3.19) | 1.60 (1.07–2.37) | 1.18 (0.79–1.77) | 1.64 (0.93–2.90) | 1.34 (0.75–2.40) | 1.03 (0.57–1.87) |
| | ≥70 | 1.13 (0.66–1.94) | 0.97 (0.58–1.64) | 1.01 (0.60–1.72) | 0.86 (0.39–1.88) | 0.66 (0.31–1.41) | 0.67 (0.30–1.47) |
| **All arterial events** | <70 | 2.21 (2.12–2.31) | 1.31 (1.25–1.37) | 0.94 (0.90–0.99) | 1.45 (1.35–1.55) | 1.08 (1.01–1.16) | 0.88 (0.82–0.95) |
| | ≥70 | 0.90 (0.87–0.93) | 0.69 (0.66–0.71) | 0.72 (0.70–0.75) | 1.15 (1.10–1.20) | 0.69 (0.66–0.72) | 0.71 (0.68–0.74) |
| MI | <70 | 2.24 (2.12–2.37) | 1.39 (1.30–1.47) | 0.94 (0.88–1.00) | 1.42 (1.29–1.55) | 1.14 (1.04–1.26) | 0.88 (0.80–0.97) |
| | ≥70 | 0.93 (0.88–0.98) | 0.74 (0.71–0.78) | 0.74 (0.70–0.78) | 1.19 (1.12–1.27) | 0.78 (0.73–0.83) | 0.75 (0.70–0.80) |
| Ischaemic stroke[4] | <70 | 2.18 (2.05–2.33) | 1.23 (1.14–1.31) | 0.90 (0.83–0.97) | 1.46 (1.32–1.62) | 1.00 (0.89–1.11) | 0.80 (0.72–0.90) |
| | ≥70 | 0.88 (0.84–0.93) | 0.65 (0.63–0.68) | 0.71 (0.68–0.75) | 1.13 (1.06–1.19) | 0.64 (0.60–0.67) | 0.69 (0.66–0.73) |
| Other arterial | <70 | 1.72 (1.28–2.31) | 0.95 (0.70–1.30) | 0.69 (0.50–0.94) | 1.63 (1.10–2.43) | 1.13 (0.74–1.72) | 0.85 (0.55–1.32) |
| | ≥70 | 0.59 (0.46–0.76) | 0.51 (0.40–0.65) | 0.52 (0.41–0.67) | 0.64 (0.47–0.85) | 0.47 (0.35–0.62) | 0.46 (0.34–0.62) |
| **Haematological events** | | | | | | | |
| Any thrombocytopenia | <70 | 2.10 (1.60–2.77) | 1.48 (1.11–1.97) | 1.00 (0.75–1.34) | 2.03 (1.42–2.92) | 1.54 (1.07–2.23) | 0.97 (0.66–1.41) |
| | ≥70 | 0.85 (0.64–1.13) | 0.72 (0.54–0.95) | 0.68 (0.51–0.90) | 1.22 (0.87–1.71) | 0.90 (0.64–1.25) | 0.79 (0.56–1.12) |
| **Other events** | | | | | | | |
| Haemorrhagic stroke[5] | <70 | 1.39 (1.13–1.71) | 0.83 (0.67–1.03) | 0.77 (0.62–0.96) | 1.31 (1.00–1.73) | 0.90 (0.68–1.19) | 0.86 (0.65–1.15) |
| | ≥70 | 0.80 (0.69–0.91) | 0.59 (0.52–0.68) | 0.65 (0.57–0.74) | 1.28 (1.09–1.50) | 0.72 (0.62–0.84) | 0.80 (0.68–0.94) |
| Mesenteric thrombosis | <70 | 1.61 (1.10–2.36) | 0.84 (0.56–1.26) | 0.65 (0.43–0.97) | 1.77 (1.11–2.84) | 1.08 (0.66–1.75) | 0.83 (0.50–1.37) |
| | ≥70 | 0.72 (0.57–0.90) | 0.54 (0.43–0.67) | 0.54 (0.44–0.68) | 1.06 (0.81–1.37) | 0.61 (0.48–0.78) | 0.59 (0.45–0.76) |
| Lower limb fracture | <70 | 1.44 (1.31–1.59) | 1.03 (0.93–1.14) | 0.93 (0.84–1.02) | 1.24 (1.08–1.42) | 0.95 (0.83–1.09) | 0.89 (0.78–1.03) |
| | ≥70 | 0.99 (0.93–1.05) | 0.65 (0.62–0.69) | 0.69 (0.66–0.73) | 1.48 (1.38–1.59) | 0.67 (0.63–0.72) | 0.71 (0.66–0.76) |
| Death | <70 | 0.74 (0.69–0.79) | 0.35 (0.32–0.37) | 0.24 (0.22–0.26) | 0.91 (0.84–1.00) | 0.52 (0.48–0.57) | 0.33 (0.30–0.37) |
| | ≥70 | 0.22 (0.22–0.23) | 0.16 (0.16–0.16) | 0.19 (0.19–0.20) | 0.30 (0.29–0.31) | 0.16 (0.16–0.16) | 0.19 (0.18–0.19) |

[1]Stratifed by region.

[2]Adjusted for age, age$^2$, sex, and stratified by region.

[3]Adjusted for age, age$^2$, sex, ethnicity, deprivation, smoking, previous diagnosis of cancer, number of unique diseases in the last year, surgery in the last year, obesity, liver disease; additionally for women aged <70, history of depression and total number of types of medication (by BNF chapters); additionally for men aged <70, history of SARS-CoV2 infection and thrombocytopenia; and additionally for men and women aged ≥70, chronic kidney disease and dementia. Venous events adjusted for: anticoagulant medication, combined oral contraceptive medication, hormone replacement therapy medication, history of PE or DVT, and history of coronavirus infection. Arterial thromboses adjusted for: diabetes, hypertension, smoking, antiplatelet medication, blood pressure lowering medication, lipid lowering medication, anticoagulant medication, history of stroke, and history of MI. All analyses stratified by region.

[4]Ischaemic stroke: including ischaemic stroke, stroke of uncertain cause, and spinal stroke.

[5]Hemorrhagic stroke: including intracranial haemorrhage and subarachnoid haemorrhage.

BNF, British National Formulary; CI, confidence interval; DVT, deep vein thrombosis; HR, hazard ratio; ICVT, intracranial venous thrombosis; MI, myocardial infarction; PE, pulmonary embolism; SARS-CoV-2, Severe Acute Respiratory Syndrome Coronavirus 2.

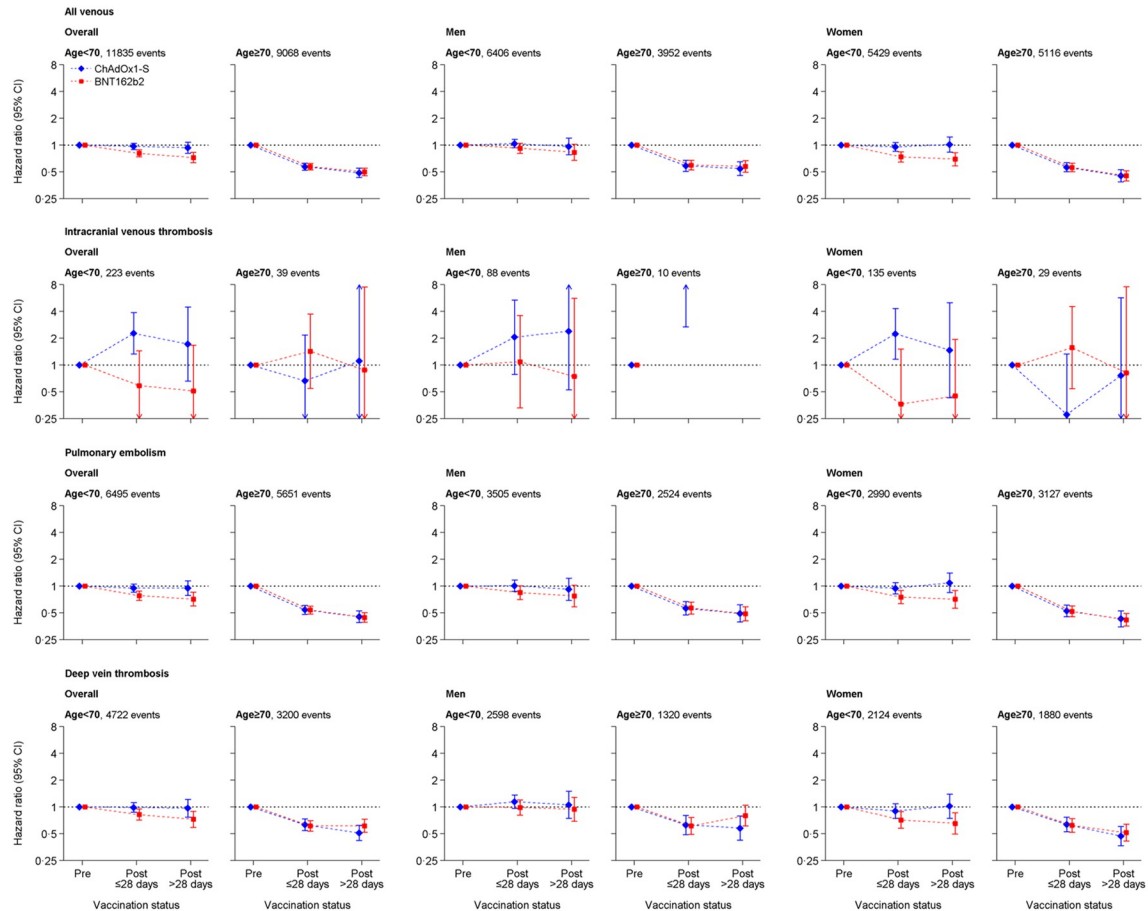

**Fig 1. aHRs for all venous thromboses, intracranial venous thromboses, PE, and deep vein thromboses after ChAdOx1-S or BNT162b2 vaccine.** aHR, adjusted HR; CI, confidence interval; PE, pulmonary embolism.

rates of arterial or venous thromboses after vaccination with ChAdOx1-S or BNT162b2 (S2 Table).

## Sensitivity analyses

When using a less restrictive outcome definition (outcomes recorded as primary or secondary reason for admission or death), aHRs were consistent with those from analyses of outcomes in the primary position (S3 Fig). Similarly, aHRs for fatal outcomes were similar to HRs for fatal or nonfatal outcomes (S4 Fig). There were very few people for whom outcomes in the primary position were recorded as well as thrombocytopenia in any position in the same hospital admission or death record (after ChAdOx1-S and BNT162b, respectively: ICVT 7 and 0; all arterial thromboses 47 and 58; all venous thromboses 37 and 26).

## Intracranial venous thrombosis

Characteristics were similar among patients who developed ICVT while unvaccinated, after ChAdOx1 and after BNT162b. Most were women (60% unvaccinated, 67% ChAdOx1, 79% BNT162b), of white ethnicity (75% unvaccinated, 92% ChAdOx1, 100% BNT162b), and had no prior comorbidities (71% unvaccinated, 50% ChAdOx1, 50% BNT162b). Patients with

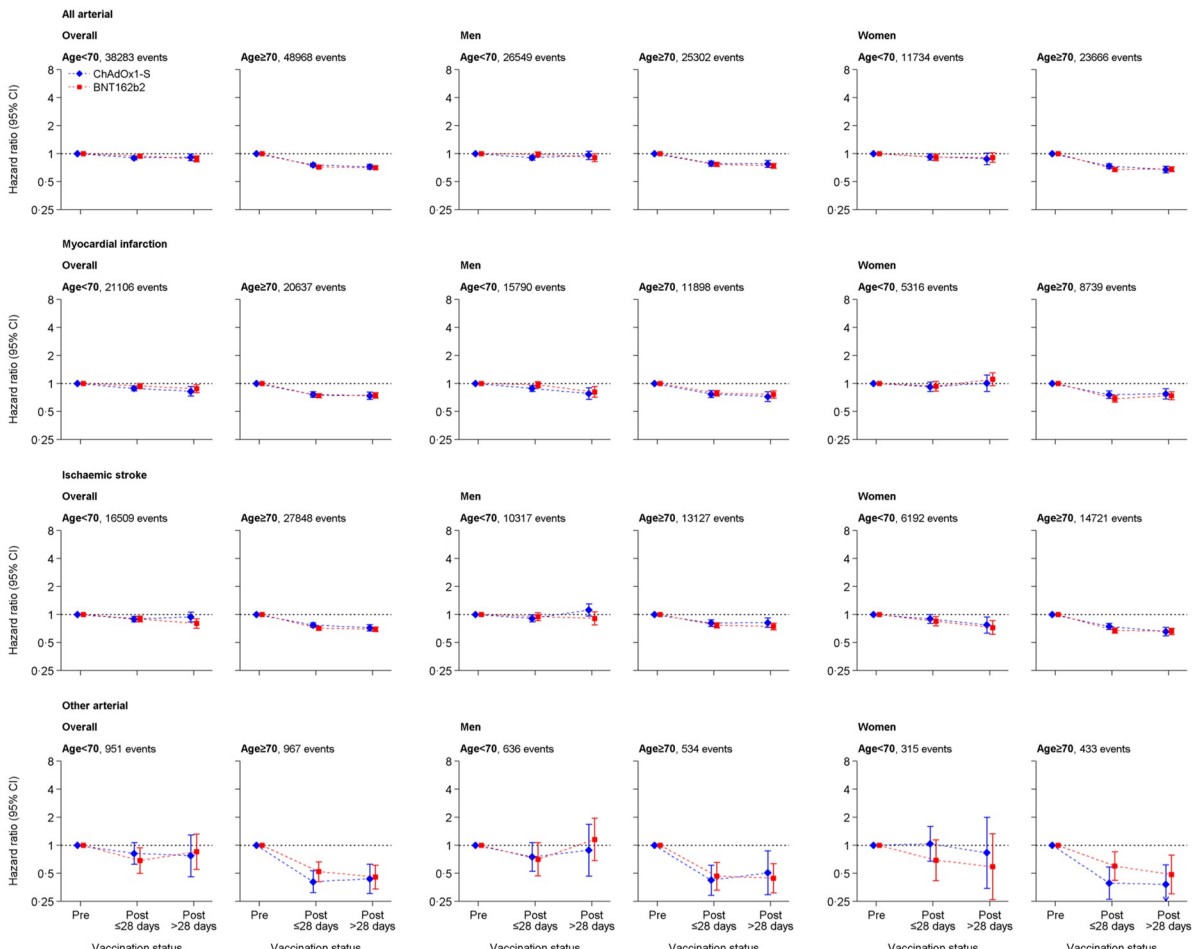

**Fig 2.** aHRs for all arterial thromboses, MI, ischaemic stroke, and other arterial thromboses after ChAdOx1-S or BNT162b2 vaccine. aHR, adjusted HR; CI, confidence interval; MI, myocardial infarction.

postvaccination ICVT were older than those with unvaccinated ICVT. Postvaccination, no patients had a recorded history of thrombophilia, and fewer than 5% had a recorded dispensed prescription of oral contraceptive or HRT (S3 Table).

Applying the aHR for ICVT 1 to 28 days after ChAdOx1 to the monthly incidence of ICVT in 2019, the excess risk of ICVT in the month after vaccination with ChAdOx1 among those aged 19 to 30, 31 to 39, 40 to 49, 50 to 59, and 60 to 69 years was estimated to be, respectively: 0.9, 1.1, 1.5, 1.5, and 1.6 per million vaccinated in men and 3.0, 2.7, 2.0, 1.6, and 1.5 per million vaccinated in women.

## Discussion

In this cohort study, which included almost all adults alive in England at the start of the national COVID-19 vaccination programme, vaccination with ChAdOx1-S, but not BNT162b2, was associated with approximately 2-fold higher rates of ICVT or hospitalisations due to thrombocytopenia in people aged under 70 years, after adjusting for a comprehensive range of demographic characteristics and comorbidities. The corresponding absolute increases in the risk of these events were very small. ChAdOx1-S and BNT162b2 were associated with

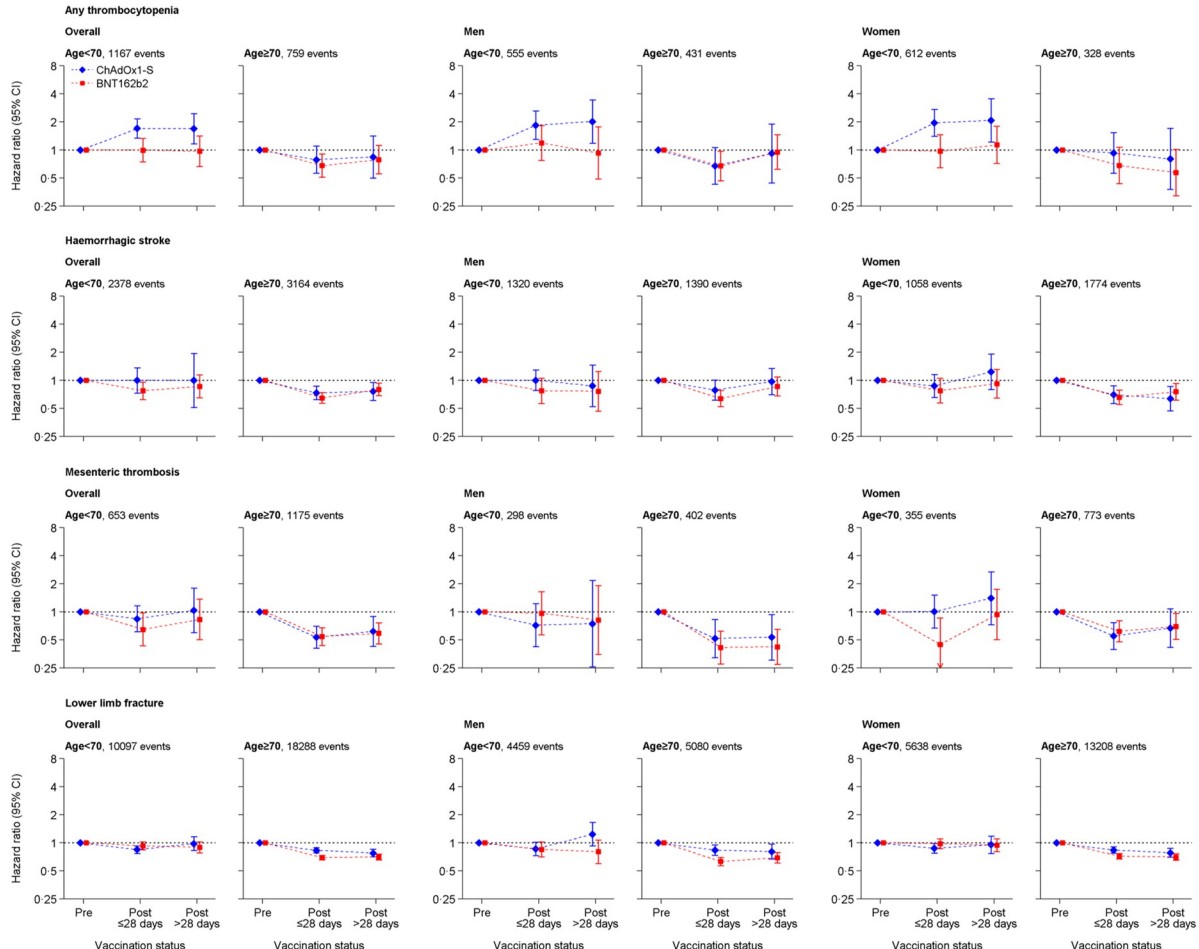

**Fig 3. aHRs for thrombocytopenia, haemorrhagic stroke, mesenteric thrombosis, and lower limb fracture after ChAdOx1-S or BNT162b2 vaccine.** CI, confidence interval.

lower hazards of major vascular events after vaccination in those aged ≥70 years. Post hoc analyses stratified by age suggest that in people aged <50 years, rates of venous thromboses were higher 1 to 28 days after vaccination with ChAdOx1-S, and that rates of arterial thrombotic events were higher 1 to 28 days after vaccination with ChAdOx1-S or BNT162b2, compared with rates in unvaccinated people.

The large population analysed adds substantial precision to estimated associations of thromboses with vaccination, which is of particular importance for very rare events and associations in subgroups. The extensive data linkages across different healthcare settings enabled adjustment for a wide range of confounding variables. The analysed population is slightly larger than the midyear estimate of the 2020 English population aged ≥18 (44,456,850) that excluded short-term residents and students, and modelled international migrant numbers [12].

Our findings are generally consistent with other studies. Smaller Danish and Norwegian electronic health record studies reported a small excess of intracranial haemorrhage (2 per 100,000 1 to 28 days after ChAdOx1-S), cerebral venous thrombosis (3 per 100,000), and other venous thrombosis (2 per 100,000) [13]. The randomised trials of ChAdOx1-S reported no venous thrombotic events among 8,597 participants who received the vaccine versus 2 among

8,581who received placebo [14]. A Scottish study found a risk of 1.13 (0.62 to 1.63) cases of idiopathic thrombocytopenic purpura per 100,000 after ChAdOx1-S. There was no clear evidence on other outcomes, although the number of postvaccine events was small [15]. A study with English vaccine data linked to other health records used a self-controlled case series (SCCS) analysis and found higher incidence rates following ChAdOx1 of: thrombocytopenia 8 to 14 days and 22 to 28 days postvaccine; of all venous thromboembolism 8 to 14 days postvaccine; and ICVT 8 to 14 days postvaccine (incidence rate ratio 4.01, 95% CI: 2.08 to 7.71); and following BNT162b2 of ICVT 15 to 21 days postvaccine (3.58, 1.39 to 9.27) and of ischaemic stroke 15 to 21 days postvaccine (1.12, 1.04 to 1.20) [16]. We did not find evidence of higher rates of any event in the whole population following BNT162b2. Differences between the results of this SCCS study and our findings may have arisen due to the following: differences in outcome data sources and definitions; inclusion in the SCCS study of outcomes recorded after the potential association of unusual thromboses and thrombocytopenia with COVID-19 vaccination was widely known, introducing possible diagnostic bias; and censoring by death in the SCCS study, a known limitation of the SCCS design for potentially fatal outcomes. Reporting of ICVT cases to the UK Medicines and Healthcare Products Regulatory Agency (MHRA) identified more cases of ICVT with thrombocytopenia after ChAdOx1-S than the present study (44 up to 31 March 2021) [17].

Healthcare systems planning to use ChAdOx1-S should balance the very small harms against the known benefits of the vaccine. For older populations, who are most vulnerable to COVID-19, we found no evidence of increased risk of any event with ChAdOx1-S. In younger populations, who have a lower morbidity and mortality due to COVID-19, other available vaccines might be prioritised, especially when the risk of COVID-19 is otherwise low.

Our study has several limitations. First, identification of exposures, covariates, and outcomes relies on the accuracy of data collected during routine healthcare. Additional data on results of laboratory and radiology investigations would have improved diagnostic coding, particularly for ascertainment of thrombocytopenia. Second, people who were not registered with an NHS GP (for example, the homeless, recent immigrants, those using only private healthcare, and those not eligible for NHS care) or who opted out of their data being provided to NHS Digital were excluded. Third, follow-up ended on 18 March 2021, but a small number of events that occurred before this date may have been excluded because they had yet to be coded or the people affected were still in hospital. Analyses after this date will likely lead to overestimation of associations, because speciality societies recommended further investigations of mild symptoms in vaccinated populations. Fourth, our primary outcome used the primary reason for death or hospital admission, which improves the positive predictive value but may lead to an underestimation of incidence. We believe this is necessary because historical nonincident events are frequently recorded in secondary positions. Analyses of events recorded in any position as fatal within 28 days were consistent with the primary analyses. Fifth, we did not address time-varying confounding, which can occur when factors, which vary during follow-up, such as admission to hospital, predict both vaccination and outcomes of interest. Sixth, comparison of adjusted and unadjusted associations suggests that studies that do not adjust for a comprehensive range of potential confounders will overestimate the thrombotic effects of vaccination. However, adjusted associations in this study may still be biased by unmeasured confounding by patient characteristics that predict both vaccination and thromboses and that are difficult to ascertain in electronic health records. Examples include general health at the time of vaccination, and people at higher risk of thromboses (for example, with end-stage diseases) [18]. There were slightly lower postvaccination rates of the "negative control" outcome lower limb fracture in those aged ≥70 years, which are likely due to unmeasured confounding. This

implies that inverse associations of vaccination with thrombotic outcomes in that age group may also be attributable in part to unmeasured confounding.

Associations of vaccination with thromboses varied with age. This may be because in older people, small increases in the risk of major thrombotic events after vaccination with ChAdOx1-S were more than offset by reductions in major thrombotic events (particularly PE) subsequent to COVID-19. By contrast, in younger people, any increase in risk associated with ChAdOx1-S vaccine is less likely to be offset by a lower risk of COVID-associated thrombosis, because the chance of severe COVID disease is lower in younger than older individuals.

Further analyses assessing the effects of other vaccines and the effects of second doses of these vaccines on thrombotic, neurological, and cardiac complications will be important to inform vaccination programmes. Access to data from radiology or laboratory systems (which in the UK will rely on regional rather than national data collection systems) will allow more comprehensive case ascertainment and more granular phenotyping. Such efforts are currently underway across the UK.

In summary, in this study, we observed increases in rates of ICVT and thrombocytopenia after ChAdOx1-S vaccination in adults aged <70 years that were small compared with its previously reported effect in reducing COVID-19 morbidity and mortality, although more precise estimates for adults <40 years are needed. For people aged ≥70 years, rates of arterial or venous thrombotic events were generally lower after either vaccine.

## Supporting information

**S1 Protocol. Prespecified protocol.**
(PDF)

**S1 RECORD Checklist. The RECORD statement—checklist of items, extended from the STROBE statement, that should be reported in observational studies using routinely collected health data.**
(DOCX)

**S1 Table.** Estimates for (a) age-stratified risk of venous and arterial events during follow-up; (b) age-stratified incidence of events pre- and post-first ChAdOx1-S; (c) age-stratified incidence of events pre- and post-first BNT162b2; (d) age- and sex-stratified HR of events 1–28 and >28 days post-ChAdOx1-S vaccine; and (e) age- and sex-stratified HR of events 1–28 and >28 days post-ChAdOx1-S vaccine BNT162b2 vaccine. HR, hazard ratio.
(PDF)

**S2 Table. Prespecified subgroup estimates for (a) venous and (b) arterial events.**
(PDF)

**S3 Table. Characteristics of patient who had an ICVT event before and after vaccination.**
Only percentage are presented, as disclosure control does not allow numbers <5 to be released. ICVT, intracranial venous thrombosis.
(PDF)

**S1 Fig. Cumulative frequency of vaccines of different types during follow-up.**
(PNG)

**S2 Fig. aHRs for portal vein thrombosis, other venous events, and death after ChAdOx1-S or BNT162b2 vaccine.** aHR, adjusted HR; CI, confidence interval.
(PDF)

**S3 Fig.** HRs for major (A) arterial and (B) venous thrombotic events and (C) haematological events, other events and lower limb fractures recorded in any position in EHR. CI, confidence interval; EHR, electronic health records; HR, hazard ratio.
(PDF)

**S4 Fig.** HRs for major (A) arterial and (B) venous thrombotic events and (C) haematological events, other events and fractures recorded in death record or in hospital record in first position followed by death <28 days. CI, confidence interval; HR, hazard ratio.
(PDF)

## Acknowledgments

This work uses data provided by patients and collected by the NHS as part of their care and support. We would also like to acknowledge all data providers who make anonymised data available for research.

## Author Contributions

**Conceptualization:** William N. Whiteley, Jennifer A. Cooper, Spencer Keene, Venexia Walker, Ashley Akbari, Emanuele Di Angelantonio, Spiros Denaxas, Angela Wood, Jonathan A. C. Sterne, Cathie Sudlow.

**Data curation:** Jennifer A. Cooper, Thomas Bolton, Spencer Keene, Venexia Walker, Rachel Denholm, Ashley Akbari, Efosa Omigie, Sam Hollings, Spiros Denaxas.

**Formal analysis:** Samantha Ip, Jennifer A. Cooper, Thomas Bolton, Spencer Keene, Venexia Walker, Efosa Omigie, Sam Hollings, Angela Wood.

**Funding acquisition:** William N. Whiteley, Ashley Akbari, Angela Wood, Jonathan A. C. Sterne, Cathie Sudlow.

**Methodology:** William N. Whiteley, Samantha Ip, Venexia Walker, Spiros Denaxas, Angela Wood, Jonathan A. C. Sterne, Cathie Sudlow.

**Project administration:** William N. Whiteley, Angela Wood, Jonathan A. C. Sterne, Cathie Sudlow.

**Software:** Samantha Ip, Jennifer A. Cooper, Spencer Keene, Venexia Walker, Sam Hollings.

**Supervision:** William N. Whiteley, Angela Wood, Jonathan A. C. Sterne, Cathie Sudlow.

**Visualization:** Samantha Ip, Thomas Bolton, Venexia Walker.

**Writing – original draft:** William N. Whiteley, Jonathan A. C. Sterne.

**Writing – review & editing:** Samantha Ip, Jennifer A. Cooper, Thomas Bolton, Spencer Keene, Venexia Walker, Rachel Denholm, Ashley Akbari, Emanuele Di Angelantonio, Spiros Denaxas, Angela Wood, Jonathan A. C. Sterne, Cathie Sudlow.

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
