## [Editor Report · Decision Letter 0]

19 Aug 2021

Dear Dr Whiteley, 

Thank you for submitting your manuscript entitled "Association of COVID-19 vaccines ChAdOx1 and BNT162b2 with major venous, arterial, and thrombocytopenic events: whole population cohort study in 46 million adults in England" for consideration by PLOS Medicine.

Your manuscript has now been evaluated by the PLOS Medicine editorial staff and I am writing to let you know that we would like to send your submission out for external peer review.

Please re-submit your manuscript within two working days, i.e. by Aug 23 2021 11:59PM.

Kind regards,

Louise Gaynor-Brook, MBBS PhD

Senior Editor

PLOS Medicine

---

## [Decision Letter · Decision Letter 1]

22 Sep 2021

Dear Dr. Whiteley,

Thank you very much for submitting your manuscript "Association of COVID-19 vaccines ChAdOx1 and BNT162b2 with major venous, arterial, and thrombocytopenic events: whole population cohort study in 46 million adults in England" (PMEDICINE-D-21-03568R1) for consideration at PLOS Medicine. 

Your paper was evaluated by three independent reviewers, including a statistical reviewer, and was discussed among all the editors here and with an academic editor with relevant expertise. The reviews are appended at the bottom of this email and any accompanying reviewer attachments can be seen via the link below:

[LINK]

In light of these reviews, I am afraid that we will not be able to accept the manuscript for publication in the journal in its current form, but we would like to consider a revised version that addresses the reviewers' and editors' comments. Obviously we cannot make any decision about publication until we have seen the revised manuscript and your response, and we plan to seek re-review by one or more of the reviewers. 

We expect to receive your revised manuscript by Oct 13 2021 11:59PM. Please email us (plosmedicine@plos.org) if you have any questions or concerns.

We look forward to receiving your revised manuscript. 

Sincerely,

Louise Gaynor-Brook, MBBS PhD

Associate Editor 

PLOS Medicine

plosmedicine.org

Comments from the Academic Editor:

I wonder why the authors didn't add a self controlled case series design as a kind of sensitivity analysis.

General comments:

Throughout the paper, please adapt reference call-outs to the following style: "... intracerebral haemorrhage [1,3,4]." (noting the absence of spaces within the square brackets).

Title: Please revise your title according to PLOS Medicine's style. We suggest “Association of COVID-19 vaccines ChAdOx1 and BNT162b2 with major venous, arterial, or thrombocytopenic events: A population-based cohort study of 46 million adults in England” or similar

Abstract:

Please combine the Methods and Findings sections into one section, “Methods and Findings”.

Abstract Background: Provide expand on the context of why your study is important. The final sentence should clearly state the study question.

Abstract Methods and Findings:

Please provide brief demographic details of the study population (e.g. sex, age, ethnicity, etc)

Please specify the study design i.e. population-based cohort study

Please clarify what is meant by ‘during follow up’

Please provide the incidence rates and/or absolute risks of relevant outcomes, not just aHRs

Please include more detail on the important dependent variables that are adjusted for in the analyses.

Lines 74, 76 - please clarify whether these are HRs or aHRs

In the last sentence of the Abstract Methods and Findings section, please describe 2-3 of the main limitations of the study's methodology.

Abstract Conclusions:

Please begin your Abstract Conclusions with "In this study, we observed ..." or similar, to summarize the main findings from your study, without overstating your conclusions. Please emphasize what is new and address the implications of your study, being careful to avoid assertions of primacy. 

Author Summary:

In the final bullet point of ‘What Do These Findings Mean?’, please describe the main limitations of the study in non-technical language.

Introduction:

Please indicate whether your study is novel, being careful to temper assertions of primacy. 

Methods:

Please state that your study had a prospective protocol / analysis plan early in the Methods section; please include the relevant prospectively written document with your revised manuscript as a Supporting Information file to be published alongside your study. A legend for this file should be included at the end of your manuscript. Changes in the analysis-- including those made in response to peer review comments-- should be identified as such in the Methods section of the paper, with rationale. If a reported analysis was performed based on an interesting but unanticipated pattern in the data, please be clear that the analysis was data-driven.

Thank you for providing a RECORD checklist. Please add the following statement, or similar, to the Methods: "This study is reported as per the Strengthening the REporting of studies Conducted using Observational Routinely-collected Data (RECORD) guideline (S1 Checklist)." When completing the checklist, please use section and paragraph numbers, rather than page numbers which will likely no longer correspond to the appropriate sections after copy-editing.

Line 159 - please refer to a specific file in your supporting information e.g. S1 Table 

Results: 

Please define the length of follow up (eg, in mean, SD, and range).

Line 265 - please clarify which vaccination ‘post-vaccinated’ refers to 

Supplementary Table 1 is central to the understanding of the paper. Please incorporate it into the main paper. 

Where aHRs are reported, please specify the comparison group.

Discussion:

Please present and organize the Discussion as follows: a short, clear summary of the article's findings; what the study adds to existing research and where and why the results may differ from previous research; strengths and limitations of the study; implications and next steps for research, clinical practice, and/or public policy; one-paragraph conclusion.

Line 354 - please temper assertions of primacy by adding ‘to the best of our knowledge’ or similar

Figures:

Please provide Figures 1 - 3 in your revised manuscript. 

Please indicate in the figure caption the meaning of the error bars in Supplementary Figures 2-4

Supplementary Fig. 1 - Please consider avoiding the use of red and green in order to make your figure more accessible to those with colour blindness.

Tables:

Please define all abbreviations used in the table legend of each table.

Table 1 - please clarify whether this is risk per 100,000 people pre- or post-vaccination 

Please specify the variables adjusted for in the table legend of Supplementary Table 1.

When a p value is given, please specify the statistical test used to determine it in the table legend.

References:

Please ensure that journal name abbreviations match those found in the National Center for Biotechnology Information (NCBI) databases, and are appropriately formatted and capitalised.

Please also see https://journals.plos.org/plosmedicine/s/submission-guidelines#loc-references for further details on reference formatting. 

Where websites are cited, please provide date of access. 

Supplementary files: 

Please see https://journals.plos.org/plosmedicine/s/supporting-information for our supporting information guidelines. 

Comments from the reviewers:

Reviewer #1: I confine my remarks to statistical aspects of this paper. The general approach is fine and appropriate but I have a couple issues to resolve before I can recommend publication.

One is that the authors have the entire population. This makes use of p values and confidence intervals problematic. These statistics are about inference from a sample to a population, but you have the population. My preference would be to remove p values and CIs. Some statisticians disagree and say that you can posit a sort of "super population". I don't find their arguments persuasive, but I won't reject an article that uses this idea, but it has to be mentioned.

I would like more explanation of why followup was split up the way it was. Usually, categorizing continuous variables is a bad idea.

Tables: The formatting of the numbers isn't great. Either add commas or remove the last 3 digits and note that the numbers are in thousands. I prefer the latter, and then some numbers will have decimal parts and some won't. 

Peter Flom

Reviewer #2: Thank you for the opportunity to review this timely manuscript on a matter of importance to public health. The size of the source data for this study is very impressive. While I commend the authors on providing a wealth of analyses on English data, I have some concerns about the analytical approach and the presentation which I hope can be adressed. 

- You state that for computational efficiency only 10% of the almost nationwide data is used. First of all, if this is correct then this is not a nationwide study. Second, how are you ensuring that the 10% selection is random; if you select individuals with equal probability, you are selecting individuals that may not have a lot of follow-up with the same probability as individuals with complete follow-up. Third, you can only be certain that an individual is a "case" or "control" if you look at the whole follow-up period. That is, you are in some way conditioning on the future if we look at the data from a survival analysis perspective. I think you need to provide more detail on this aspect of your analysis and reassure the reader that this is a valid approach for survival data. Please also include references to where this approach has been used before. Alternatively, use another approach such as Poisson regression which should be more computational feasible for all the data. 

- You should not use any stepwise selection procedure for covariate inclusion.

- Please elaborate on your approach to absolute excess risk. Is this an observed (post-vac) vs expected (historic) analysis? Please provide 95%CI also when reporting these risks. 

- The tables are overloaded, have formatting issues and are difficult to read. 

- Why are the main results "hidden" in a supplementary table?

- I am lacking a hierarchy of results. As it stands, you are presenting all possible combinations of analyses. Do all these analytical combinations represent apriori hypotheses? 

- You include fractures as a control outcome supposedly not associated with vaccination (which is not entirely clear since fractures and frailty could be associated) and present significantly protective effects of vaccination without comment? Could this protective effect not represent unadjusted confounding/bias, which would also have biased thrombotic outcome risks towards null?

Reviewer #3: This is an interesting comprehensive analysis of the risk of venous and arterial thrombosis, or thrombocytopenia in association with ChAdOx1-S and BNT162b2 vaccination in the English adult population. I have some comments that should be addressed:

The definition of the unvaccinated group is unclear. This group includes pre-vaccination times of later vaccinated persons as well as unvaccinated persons. What was the proportion of pre-vaccination times and unvaccinated persons? How was the observation time defined in the unvaccinated population? How long was the observation time? How were the controls (unvaccinated) selected? This information needs to be provided in much more detail-

Including prevaccination times is of concern. Patients who develop VTE, ATE or thrombocytopenia would probably not get vaccinated. This introduces substantial bias.

„…the monthly incidence of ICVT in 2019.." From where did you derive this data? Please provide information in the Methods.

How did you handle missing data?

„Rates of intracranial venous thrombosis and thrombocytopenia in adults …" This should be „OR" rathern than „AND", since you did not link the two events. In light of VITT, current presentation might be misleading particularly considering the very heterogeneous definition of thrombocytopenia. Please report thrombotic events separately from thrombocytopenia and rephrase. 

Data on VTE and ATE risks in the general population are important to assess their risks in association with the vaccination. This would also be of interest for immunethrombocytopenia. However, the definition of thrombocytopenia in the present study is a mixed bag and includes a wide range of diseases associated with a low platelet count (e.g. TTP). It is unclear what the relevance of this data is. 

„…our primary outcome used the primary reason for death or hospital admission.." Are all DVT treated in the hospital in the UK? Is there the possibility to have missed events that were treated in an outpatient setting?

„In younger populations, who have a lower morbidity and mortality due to COVID-19, other available vaccines might be prioritise, especially when the risk of COVID-19 is otherwise low." Given the low absolute risk this conclusion cannot necessarily be drawn. The decision on the vaccine is influenced by other factors including availability and costs. Also, how would one define an otherwise low risk of COVID-19?

[LINK]

---

## [Decision Letter · Decision Letter 2]

24 Nov 2021

Dear Dr. Whiteley,

Thank you very much for submitting your manuscript "Association of COVID-19 vaccines ChAdOx1 and BNT162b2 with major venous, arterial, or thrombocytopenic events: A population-based cohort study of 46 million adults in England" (PMEDICINE-D-21-03568R2) for consideration at PLOS Medicine. 

Your paper was evaluated by an associate editor and discussed among all the editors here. It was also seen again by the reviewers. The reviews are appended at the bottom of this email and any accompanying reviewer attachments can be seen via the link below:

[LINK]

In light of these reviews, I am afraid that we will not be able to accept the manuscript for publication in the journal in its current form, but we would like to consider a revised version that addresses the reviewers' and editors' comments. Obviously we cannot make any decision about publication until we have seen the revised manuscript and your response, and we plan to seek re-review by one of the reviewers. 

We hope to receive your revised manuscript by Dec 15 2021 11:59PM. Please email us (plosmedicine@plos.org) if you have any questions or concerns.

We look forward to receiving your revised manuscript. 

Sincerely,

Callam Davidson (on behalf of Louise Gaynor-Brook, MBBS PhD)

Associate Editor

PLOS Medicine

plosmedicine.org

It appears a large amount of content has been duplicated in your responses to the submission form questions, please revise as appropriate. 

The lines ‘This work uses data provided by patients and collected by the NHS as part of their care and support. We would also like to acknowledge all data providers who make anonymised data available for research’ from your Financial Disclosure statement would be better placed in the Acknowledgements section in the main text.

In your Data Availability Statement, please additionally provide either a URL or email address that an interested researcher could use to apply for access to the de-identified data (the current URL only describes the project and does not detail the application process).

Please ensure citations throughout the text are in square brackets, are not in sub- or superscript, and appear before punctuation (e.g. ‘…in the absence of exposure to heparin [1].’).

In the interest of brevity, please remove the absolute incidence rates but include a sentence to the effect of ‘Substantial confounding was observed in the absolute incidence rates, but after adjustment…’.

In the interest of brevity, please shorten the list of independent variables adjusted for to ‘Analyses were performed separately for ages <70 and ≥70 years, and adjusted for age, age2, sex, ethnicity, deprivation and relevant medical history. Further covariates were selected using backward selection.’

Lines 129 – 132: please combine these two bullet points that reads ‘We used nationally collated data from electronic health records on 46 million adults, of whom 21 million were vaccinated during the study, and compared the incidence of venous and arterial events before and after the first vaccination with ChAdOx1-S and BNT162b2 COVID-19 vaccines.’

Line 179: When citing your supplementary material, please provide a specific item number (e.g. S1 etc.). See our supporting information guidelines for more details: https://journals.plos.org/plosmedicine/s/supporting-information

Line 197: As above, please cite the specific item within the supporting information (e.g. S1 protocol).

Please ensure all abbreviations are defined on first use (e.g. MI, DVT, PE are all defined in the ‘outcomes’ section currently but are used earlier in the manuscript).

In the results section, please include the relevant quantitative results when stating observations. 

Please include the absolute incidence rates in the results (around line 333).

Line 377: Please only report p-values to 3 significant figures (P<0.001 rather than P<0.0001). Please check throughout. 

Line 449: Please correct ‘prioritise’ to ‘prioritised’.

Please remove the ‘Data Sharing’, ‘Funding’, Conflicts of interest’ and ‘Contributions’ sections from the end of the main text. In the event of publication this information will be published as metadata based on your responses to the submission form questions. 

Reference 15: Please add [preprint] per our referencing guidelines (https://journals.plos.org/plosmedicine/s/submission-guidelines#loc-references) 

Comments from the reviewers:

Reviewer #1: The authors have addressed my concerns and I now recommend publication

Peter Flom

Reviewer #2: Thank you for the revision. The authors should reference, "Risk of thrombocytopenia and thromboembolism after covid-19 vaccination and SARS-CoV-2 positive testing: self-controlled case series study - PubMed (nih.gov)" which uses the same data and discuss the differences between their study and the already published study. 

My comments to the responses:

35. I do not understand why I am being lectured on IP weights from causal inference when that is not what has been done here, you do not use the probability of treatment/non-treatment in your weights, just the selection probability. Your study is still not a cohort study of 46 million. Your study is essentially a case control study. Please provide a reference for your simulation studies.

36. Stepwise selection is still not a valid approach to confounder control. How does stepwise selection remove collinearity? If too few events are your main concern, why would you use backwards selection - where you are going from a model with (too) many covariates potentially providing unstable estimates.

37. Ok.

38. Can you clarify what has changed. Table 1 and 2(a+b) seems very similar to previous editions.

39-41. Ok.

Reviewer #3: Comments well addressed

[LINK]

---

## [Decision Letter · Decision Letter 3]

14 Jan 2022

Dear Dr. Whiteley,

Thank you very much for re-submitting your manuscript "Association of COVID-19 vaccines ChAdOx1 and BNT162b2 with major venous, arterial, or thrombocytopenic events: A population-based cohort study of 46 million adults in England" (PMEDICINE-D-21-03568R3) for review by PLOS Medicine.

I have discussed the paper with my colleagues and the academic editor and it was also seen again by one reviewer. I am pleased to say that provided the remaining editorial and production issues are dealt with we are planning to accept the paper for publication in the journal.

[LINK]

We look forward to receiving the revised manuscript by Jan 21 2022 11:59PM.   

Sincerely,

Louise Gaynor-Brook, MBBS PhD

Associate Editor 

PLOS Medicine

plosmedicine.org

Requests from Editors:

General comments:

Please ensure that a space is added before the reference call-out e.g. “AstraZeneca) [1–3].”

Please be clear when HRs presented are adjusted HRs / aHRs

Abstract Methods and Findings:

Line 69 - please revise to ‘ethnicity and deprivation’

Line 71 - please define PE and DVT at first use 

Line 74 - please revise to ‘history of stroke and history of myocardial infarction’

Line 77 - please revise to ‘were <70 years of age’

Line 79 - please revise to ‘mixed ethnicity and 1.5 million’

Line 83 - please define CI at first use 

Author Summary:

Please specify the definitions of younger and older used in this study (i.e. younger than 70 years, or 70 years or older)

Please move the limitations to be the final bullet point of ‘What Do These Findings Mean?’

Methods:

Line 221 - please specify supplementary file name (S1 Protocol)

Results: 

Line 302 - Tables 1 / S1a do not appear to show that “risks of venous and arterial thromboses were higher in people with… increasing deprivation”; please clarify 

Line 331 - please revise to aHR 

Line 335 / 344 / 345 / etc - please specify whether HRs presented in the main text are adjusted HRs

Line 336 - please clarify in which table data are presented for the post-hoc analyses; this does not appear to be either of Table 3 or S1 Table (d,e) as stated. Line 371 - please clarify in which table data are presented for the analyses of <40 compared to ≥ 40 years

Lines 375-79 - please refer to the table in which these data are presented

Line 391 - it appears from Table S3 that this should be 60% unvaccinated; please clarify

Discussion:

Lines 435-7 - please revise to ‘days post-vaccine’

Line 468 - is the underlying data for these analyses provided? PLOS do not permit “data not shown”

Tables:

NB) Table under file name ‘S2 Table’ is labelled as S1 table; also for file ‘S3 Table’ / S2 table 

S2 Table - please clarify what 1 in superscript after HR (95% CI) refers to

Comments from Reviewers:

Reviewer #2: The authors have addressed my comments satisfactorily.

[LINK]

---

## [Editor Report · Decision Letter 4]

21 Jan 2022

Dear Dr Whiteley, 

On behalf of my colleagues and the Academic Editor, Prof. Suzanne Cannegieter, I am pleased to inform you that we have agreed to publish your manuscript "Association of COVID-19 vaccines ChAdOx1 and BNT162b2 with major venous, arterial, or thrombocytopenic events: A population-based cohort study of 46 million adults in England" (PMEDICINE-D-21-03568R4) in PLOS Medicine.

PRESS

Sincerely, 

Louise Gaynor-Brook, MBBS PhD 

Associate Editor 

PLOS Medicine